# Tracing Mastitis Pathogens—Epidemiological Investigations of a *Pseudomonas aeruginosa* Mastitis Outbreak in an Austrian Dairy Herd

**DOI:** 10.3390/ani11020279

**Published:** 2021-01-22

**Authors:** Bernhard Schauer, Regina Wald, Verena Urbantke, Igor Loncaric, Martina Baumgartner

**Affiliations:** 1Department of Pathobiology, Institute of Microbiology, University of Veterinary Medicine, Veterinärplatz, 1210 Vienna, Austria; 1145133@students.vetmeduni.ac.at (B.S.); igor.loncaric@vetmeduni.ac.at (I.L.); 2Department of Farm Animal and Public Health in Veterinary Medicine, University Clinic for Ruminants, University of Veterinary Medicine, Veterinärplatz 1, 1020 Vienna, Austria; regina.wald@vetmeduni.ac.at (R.W.); verena.urbantke@vetmeduni.ac.at (V.U.)

**Keywords:** *Pseudomonas aeruginosa*, mastitis, transmission route, udder cleaning microfiber towels, MLVA

## Abstract

**Simple Summary:**

*Pseudomonas aeruginosa* is an opportunistic pathogen in humans and animals. In ruminants, it is responsible for mastitis that may occur sporadically or as an outbreak within dairy herds. It can be refractory to treatment, and therefore, economic implications are high. To avoid spread within herds, it is important to identify the sources of infection or transmission ways and to implement effective prophylactic measures. Intra-mammary infections with *P. aeruginosa* can occur from both, a point source, or from a continuous exposure to a common contamination, e.g., from dirty, or soiled environment. We describe a *P. aeruginosa* outbreak in an Austrian dairy herd were genotyping of isolates helped to identify reusable udder towels as source of subclinical and mild clinical intramammary infections with *P. aeruginosa*. Bacteriological examination revealed that in nine out of 20 lactation cows at least one quarter was infected with *P. aeruginosa*. Genotyping of isolates with multiple locus variable-number-tandem repeat analysis (MLVA) revealed that all strains were closely related to each other. The same genotype was additionally confirmed in strains isolated in pure culture from a cleaning and disinfectant solution for udder towels indicating that there is a strong evidence that transmission from cow to cow occurred via those udder wipes.

**Abstract:**

The present study describes an outbreak of *Pseudomonas (P.) aeruginosa* mastitis in a 20-cow dairy herd where throughout genotyping of isolates reusable udder towels were identified as the source of infection. Sampling of cows during three herd surveys and bacteriological culturing showed that *P. aeruginosa* was isolated from nine cows with a total of 13 infected quarters. Mastitis occurred as mild clinical or subclinical infection. *P. aeruginosa* was additionally isolated from a teat disinfectant solution, containing N-(3-aminopropyl)-N-dodécylpropane-1,3-diamine 1 as active component, and microfiber towels used for pre-milking teat preparation. Disc diffusion antimicrobial resistance testing revealed that all isolates were susceptible to piperacillin, piperacillin-tazobactam, ceftazidime, cefepime, aztreonam, imipenem, meropenem, tobramycin, amikacin, and ciprofloxacin. Thirty-two isolates of milk samples and 22 randomly selected isolates of one udder towel and of the teat disinfectant solution were confirmed as *P. aeruginosa* with matrix-assisted laser desorption, ionization time-of-flight mass spectrometry (MALDI Tof MS). Isolates were further characterized with rep-PCR and randomly amplified polymorphic DNA (RAPD) as well as with multiple locus variable-number tandem repeat analysis (MLVA). Results obtained in this study suggested that one single strain was responsible for the whole outbreak. The transmission occurred throughout a contaminated teat cleaning solution as a source of infection. The farmer was advised to change udder-preparing routine and to cull infected cows.

## 1. Introduction

*Pseudomonas (P.) aeruginosa*, a Gram-negative oxidase-positive rod shaped, motile bacterium is associated with acute and chronic infections in various species. In humans, *P. aeruginosa* is considered an opportunistic pathogen responsible for high morbidity and mortality infection rates leading to pneumonia, septicemia, and necrosis in immunocompromised patients [1]. Because of high antimicrobial resistance rates, the World Health Organization [2] additionally lists it as a critical priority pathogen. In companion animals, *P. aeruginosa* has been demonstrated to be etiological cause of infections such as otitis, dermatitis, and urinary tract infections [3].

In ruminants, it has been isolated from cases of clinical and subclinical mastitis in dairy cows, sheep, and goats [4,5,6,7]. As environmental pathogens, *Pseudomonas* spp. are widely present in humid areas. Soiled bedding, manure or contaminated water, wash hoses in milking parlors, and spray nozzles represent important sources of infection [8,9]. Biofilm formation in the milking parlor and poor environmental hygiene may facilitate IMI as risk factors [10,11]. Investigations have also traced mastitis origin to contaminated intramammary antibiotic infusions under unhygienic conditions [12].

Clinical signs seem to depend on the exposure dose and range from toxic mastitis with severe symptoms of toxemia, marked swelling of the mammary gland, high body temperature, and watery milk secretions with clots or blood, to chronic non-clinical infections with elevations of somatic cell counts [13]. The most common course is initially characterized by mild mastitis that develops into a stage of chronic infection. In sheep and goats, gangrenous mastitis with high fatality rates is observed in approximately 10% of cases [14,15].

Apart from the occasional onset of intra-mammary infections (IMI), outbreaks at the herd level with severe economic implications have been described [9,12,15].

Antibiotic treatment of acute clinical either chronic subclinical cases is usually unsuccessful even if antimicrobial susceptibility testing (AST) indicates in vitro susceptibility [13]. Therefore, fast and accurate diagnostic techniques for the identification of mastitis pathogens and detection of sources of infection and transmission routes is key to prevent outbreaks at the herd level [16]. Because of the opportunistic character and the heterogeneity of *Pseudomonas* spp. in the environment, typing methods can be used to distinguish between intramammary infections from a point source or resulting from a continuous exposure to a common contamination.

We describe an outbreak of *P. aeruginosa* mastitis in an Austrian dairy farm, where molecular diagnostic techniques helped to identify a contaminated teat cleaning solution as a point source of infection.

## 2. Materials and Methods

Ethical review and approval were waived for this study, because all examinations and treatment of animals occurred as professional activities of the local veterinarian as part of his veterinary services and medical care.

### 2.1. Outbreak History

During summer 2018, an unusual and progressive increase of milk somatic cell count (MSCC) occurred in a small dairy herd in the eastern region of Austria. Quarter milk samples (*n* = 80) of 20 dairy cows were submitted in September 2018 to the diagnostic laboratory of the University Clinic for Ruminants of the University of Veterinary Medicine, Vienna, Austria, for bacteriological culturing and *P. aeruginosa* was isolated from six cows. Three weeks later a herd visit was performed to re-sample all lactating cows and to identify possible sources of infection. During this visit, information on housing type, herd size, milk yield, bedding material, feeding, milking technique and dry cow management were obtained. Furthermore, changes of SCC during the last month were recorded, and milking hygiene and procedure were evaluated.

Although treatment is rarely successful, four cows were selected for a high-dose-short duration treatment with fluoroquinolones. During a second herd survey in November 2018 all cows were re-sampled for bacteriological culturing and SCC measurement.

### 2.2. Collection of Samples

Milk and environmental samples were collected aseptically in the milking parlor before milking as previously described [5]. The first streams of milk were used for the California Mastitis Test (CMT) and discarded [17]. Teat ends were scrubbed with a cotton ball soaked in alcohol and milk was collected into sterile sample tubes. The inside surface of the teat cup liners and the short milking tubes were sampled with moisturized sterile cotton wool swabs that were inserted into the liners, withdrawn in a spiraling motion and collected into sterile tubes containing 2 mL of tryptic soy broth (TSB, Merck Millipore, Darmstadt, Germany). Additionally, 20 mL of the disinfectant solution that was prepared for teat cleaning and a soaked cotton towel were sampled. The latter was placed into a sterile vial with 20 mL of TSB. From each of the two cattle water troughs in the free stall barn, 20 mL of drinking water was collected for culturing. All samples were transported directly to the laboratory, cooled overnight, and processed the next day.

### 2.3. Bacteriological Examination of Quarter Milk Samples and SCC Measurement

Ten microliters of each milk sample was plated onto Columbia blood agar with 5% sheep blood (Oxoid, Ltd., Basingstoke, UK) and incubated at 37 °C for 24 and 48 h under aerobic conditions. Isolates were evaluated based on their phenotypical and biochemical criteria. Isolates were identified based on their phenotypical and biochemical properties as suggested by the National Mastitis council (NMC) [18]. Gram-negative and oxidase-positive bacilli, growing in pure culture in pigmented and mucoid colonies with a characteristic odor were identified as presumptive *P. aeruginosa.* In cases where MSCC was elevated (positive CMT or SCC >200 cells per µL) but culturing resulted in no growth after an incubation time of 48 h, an enrichment step was performed: 100 µL of milk was incubated overnight at 37 °C in 2 mL of TSB and samples were re-plated. One phenotypically as *P. aeruginosa* confirmed isolate from each milk sample was selected and stored at −80 °C in a 15% glycerol solution for confirmation and molecular strain typing. MSCC was measured in the milk of infected glands with the DCC Cell Counter (De Laval, Tumba, Sweden).

### 2.4. Antimicrobial Resistance Testing (AMR)

Antimicrobial resistance testing was carried out by agar disk diffusion according to the Clinical and Laboratory Standards Institute (CLSI, Wayne, PA, USA). Following antimicrobial agents were tested: piperacillin (PIP 100 µg), piperacillin-tazobactam (TZP 100/10 µg), ceftazidime (CAZ 30 µg), cefepime (FEP 30 µg), aztreonam (ATM 30 µg), imipenem (IPM 10 µg), meropenem (MEM 10 µg), gentamicin (GEN 10 µg), tobramycin (TOB 10 µg), amikacin (AMK 30 µg), and ciprofloxacin (CIP 5 µg) (all from Becton Dickinson, Heidelberg, Germany). Zone diameters and were interpreted according to CLSI [19].

### 2.5. Bacteriological Culturing of Environmental Samples

Teat disinfectant solution, swabs, the cotton towel (in TSB) and the water samples were incubated for 4 h at 37 °C [5]. Hundred microliters of a 1:10, 1:100, and 1:1000 dilution in TSB was plated onto Columbia agar and incubated at 37 °C. If growth of *P. aeruginosa* was detected in an environmental sample, eleven single colonies were picked up randomly from each plate and stored at −80 °C until further characterization and assessment of strain diversity.

### 2.6. Confirmation and Genotyping of P. aeruginosa

A total of 54 *P. aeruginosa* isolates (32 from IMI, 11 from disinfectant solution and 11 from the udder towel) were re-cultivated overnight on BD™ Columbia III Agar supplemented with 5% Sheep Blood (Becton Dickinson, Heidelberg, Germany). Isolates showing the typical colony morphology of *P. aeruginosa* were then identified to the species level by matrix-assisted laser desorption/ionization-time-of-flight mass spectrometry (MALDI-TOF MS) (Bruker Daltonik, Heidelberg, Germany). A MALDI-TOF score ≥2.00 was used as a threshold for reliable identification. For DNA extraction, the same *P. aeruginosa* colonies as tested with MALDI-TOF were cultivated on BD Columbia III Agar supplemented with 5% sheep blood overnight. DNA was extracted using the UltraClean Microbial DNA Isolation Kit (MoBio Laboratories, Dianova, Hamburg, Germany). DNA was not quantified. For further confirmation of isolates, a species-specific simplex PCR was performed with the primer pair sssF and sssR [20]. Enterobacterial repetitive intergenic consensus (ERIC), repetitive extragenic palindromic (REP)- and BOX-PCR techniques (collectively known as rep-PCR) were performed using the primer ERIC1R and ERIC2 [21,22]. In addition, randomly amplified polymorphic DNA (RAPD) (primer 268, 272, H12, AP7) was performed [21,22,23,24]. A multiple-locus variable-number tandem repeat analysis (MLVA) using fifteen primer pairs (Table 1) was performed for the further distinction of the *P. aeruginosa* isolates [25].

## 3. Results

Farm characteristics and clinical examination. The herd comprised 20 lactating Simmental cows (a traditional dual-purpose breed) housed in a free-stall barn with cubicles bedded with straw-manure-mattresses. Feed ration consisted of grass and maize silage supplemented with concentrates. Average milk production was 24 kg per cow and day. The number of cows with a SCC >1 million/mL increased from one cow in April 2018, up to six in august, while number of cows with a SCC <200,000/mL decreased from fourteen to five animals. Dairy cows were milked twice a day in a tandem-milking parlor. Pre-milking hygiene procedures consisted of fore stripping (performed irregularly by stripping on the parlor’s floor) and teat cleaning with one cotton towel per cow. Those towels were hand washed after milking and soaked into freshwater with added disinfectant (Dermisan Plus, Hypred, Bornheim, Germany) between two milking times. Teat skin was evaluated as being clean but still wet when teat cups were attached. A barrier dip with lactic acid as disinfectant component (LactiFence, DeLaval, Tumba, Sweden) was used for post-dipping. Further background information on the farm is summarized in Table 2.

MSCC as determined with the California Mastitis Test was highly elevated in almost one quarter for each of the nine cows. Milk secretions of three quarters showed the presence of flakes. Mild clinical symptoms (e.g., flakes) or subclinical course of the disease (elevated MSCC) characterized all IMI, while acute cases with marked swelling, high body temperature, watery milk with clots as described for severe *Ps. aeruginosa* mastitis [6] were not observed. At the herd level, mastitis occurred as a chronic and progressive infection, characterized by high MSCC and decreased milk yield. Figure 1 shows evaluation of the MSCC increase during summer 2018.

### 3.1. Bacteriological Examination and Molecular Strain Typing

Bacteriological examination of aseptically collected milk samples revealed that nine out of 20 cows were infected with *P. aeruginosa.* The latter was isolated from 17 quarter-milk samples during the whole study period (see Table 3). All isolates were Gram-negative and oxidase-positive rods, showing typical “grape-like” odor, were beta-hemolytic with mucoid metallic growing pattern on SBA, they were non-lactose fermenters on Mac Conkey agar (Oxoid) and grew in yellow-green, blue-green, reddish, or brownish colonies on Müller Hinton agar (Oxoid) as shown in Figure 2.

Resampling of cows during the following herd visits confirmed IMI with *P. aeruginosa*. Quarter milk samples of two formerly positive diagnosed cows were negative although SCC was still highly elevated in quarter milk samples. One cow was newly infected. A third herd survey carried out 1 month later proved treatment failure in three out of four cows: two cows were still bacteriologically positive for *P. aeruginosa*, a third cow was bacteriologically negative, but milk secretions were changed and SCC was highly elevated, indicating failure of clinical and cytological cure (Table 3). Overall, 32 *P. aeruginosa* isolates originating from nine cows with IMI during the three surveys were stored for further examinations.

Regarding the environmental samples two specimens yielded predominant bacterial growth of *P. aeruginosa*; from the sample of the microfiber towel *P. aeruginosa* was isolated in pure culture in a concentration of 1.27 × 10^5^ cfu/mL. While specimens of the dip cup and the post teat dip were bacteriologically negative, culturing of the water sample from the cattle watering troughs and of swabs of the teat cups and the short milking tube yielded mixed bacterial growth but was negative for *P. aeruginosa.* Beside *P. aeruginosa,* the following intramammary infections (IMI) were detected: Non-Aureus Staphylococci (NAS *n* = 2), *Enterobacter sp.* (*n* = 1), *Sc. uberis* (*n* = 1), *Sc. dysgalactiae* (*n* = 1).

### 3.2. Interventions

As the most important preventive measure, the farmer was instructed to change the pre-milking routine and clean teats with dairy paper towels (dry or in combination with a disinfectant containing other active substances).

AMR testing revealed that all strains were susceptible for ciprofloxacin. In Europe, enrofloxacin is labeled for mastitis therapy in dairy cows. Treatment of *Pseudomonas* mastitis is rarely effective. As a request of the farmer, four selected cows received a high-dosage short-duration treatment with fluoroquinolones as described for *P. aeruginosa* infections in dogs [26]. Enrofloxacin (Baytril 10%, Bayer, Leverkusen, Germany) was applicated at a dose of 10 mg/kg by slow intravenous injection as the initial treatment, followed by subcutaneous administration of 5 mg/kg/day for two days. BE confirmed that treatment of selected cows was not effective (except for one cow, which was bacteriologically negative and clinically cured), so that all affected cows were culled.

## 4. Discussion

Epidemiological investigations aim to identify sources and transmission routes of pathogens and to elucidate our understanding of mechanisms of host-adaption and disease causation [16]. In this study genotyping methods were applied to identify the source of intramammary infections with *P. aeruginosa* in a dairy herd. A widely used technique is pulsed-field gel electrophoresis (PFGE), which is expensive and time-consuming compared to other techniques but is often considered gold standard. Another technique, which is faster and cheaper, is repetitive-element-based PCR such as MLVA. MLVA proved to be a reliable method with a high discriminatory power, inexpensive, and comparable between laboratories [25]. Furthermore, MLVA is a superior technique for investigating disease outbreaks compared to PFGE because with MLVA, mutations can be detected [27]. Randomly amplified polymorphic DNA PCR such as ERIC, rep, and BOX are widely used genotyping methods. They are comparably cheap, fast, and, if used together, they exhibit high discriminatory power [28].

Although all isolates belonged to the same MLVA type, they differed in their phenotypes regarding colony morphology, e.g., color of colonies. Pigmentation of *P. aeruginosa* is the result of production of different pigments such as pyoverdine, pyocyanin, pyomelanin, and pyorubin [29,30]. Biosynthesis of each pigment is influenced by genetic arrangements and environmental conditions such as temperature, incubation time, medium composition, or oxygen tension [31]. *P. aeruginosa* pigments are important virulence factors because of their impact on iron uptake and biofilm formation [32,33].

*P. aeruginosa* is recognized as an important cause of acute and chronic diseases in humans and animals. In this study, 32 isolates originating from quarter milk samples of nine cows at different time points were genotyped and compared with isolates from an environmental source (udder wipes and teat cleaning solution). Mastitis occurred as a chronic and progressive infection at the herd-level, characterized by high SCC, decreased milk yield, and poor response to treatment. A non-clinical P. aeruginosa mastitis outbreak, where contaminated water was identified as the source of infection is reported [8]. While clinical infections usually result from single exposures to high bacterial loads, non-clinical and chronic infections are more likely to develop when the mammary gland is repeatedly exposed to smaller numbers of *P. aeruginosa* [8].

Similarly, as observed in the Irish study [9] we noticed that it was more difficult to isolate bacteria from chronic cases and that often an enrichment step was necessary to gain growth of *P. aeruginosa* from a milk sample that was formerly positive (e.g., during the first BE), or that BE resulted negative although MSCC was elevated. It is assumed, that in chronic cases this might be due to a lower concentration and intermittent shedding [9]. Another explanation for chronicity of infections might be the ability of some *P. aeruginosa* strains to form biofilms. Biofilm growing bacteria also exhibit increased tolerance against host immune mechanisms and against antibiotics or disinfectants and cause persisting inflammation and tissue damage [32].

Outbreaks of *P. aeruginosa* mastitis have been reported from several countries and epidemiological investigations revealed that in most cases a large variety of strains within and between herds can be detected [16]. In our case, the detection of *P. aeruginosa* isolates exhibiting indistinguishable molecular characteristics strongly suggested transmission within the herd. We assume that one single strain was responsible for all *P. aeruginosa* infections, persisted for several weeks (was isolated during the whole study period of three months), and was additionally isolated from the disinfectant solution microfiber towels used for teat cleaning.

The predominance of one or more single strains within herds is characteristic for contagious pathogens and transmission from cow to cow via milking machine is the most frequent spread mechanism. As *P. aeruginosa* is an environmental pathogen, this is rather unlikely. Other explanations for homogeneity of mastitis-associated strains among cows may be exposure of teat ends to one-point source and increased fitness (or resistance) of a strain [5]. There is some evidence in our study for this: Although the primary source of infection was not identified, we can assume that *P. aeruginosa* was transmitted via contaminated udder wipes. The high bacterial load and the fact that the pathogen was isolated in pure culture from towels and cleaning solution indicate that resistance of the strain toward the disinfectant solution was likely and might have been an important predisposing factor. Similarly, predominance of *P. aeruginosa* strains in connection with mastitis occurred in Irish [9] and in Dutch dairy herds [33]. Both multi-herd outbreaks were associated with one brand of disinfectant wipes used for teat disinfection before intramammary application of antibiotics. In the Irish herds, all outbreaks had been caused by the same *P. aeruginosa* strain. There are some similarities with mastitis outbreaks caused by *Serratia (S.) marcescens* in Finnish dairy farms [34]. Epidemiological characterization of isolates from two outbreaks revealed that IMIs were associated with contaminated teat dips containing a tertiary alkyl amine, n,n-bis (3-aminopropyl) dodecyl amine. Bacteriological examinations confirmed *S. marcescens* in mastitis milk and in teat dip solution and equipment, while bacteriological examination from samples from an unopened canister was negative. Puls-field-gel electrophoresis (PFGE) of isolates from both, infected cows and teat dip revealed that within both herds mastitis was caused by a predominant strain that differed between herds.

*P. aeruginosa* mastitis is challenging to treat as it is intrinsically resistant to many antimicrobials used for mastitis therapy such as ampicillin, amoxicillin, amoxicillin/clavulanic acid, cephalosporins (first, second generation), or trimethoprim [35]. Susceptibility testing revealed that all strains were susceptible to ciprofloxacin. A high-dosage short-duration treatment with enrofloxacin is recommended for urinary tract infections with *P. aeruginosa* in dogs [26]. In our case, treatment was unsuccessful, probably because of chronic course of mastitis.

## 5. Conclusions

*P. aeruginosa* is a major human pathogen; in veterinary field, it is mainly associated with sporadic diseases of the urinary tract, chronic pyodermia, and dermatitis and otitis externa in companion animals. In dairy cows, it causes intramammary infections, that generally result from a continuous exposure to a soiled environmental source. Mastitis cases within a herd are usually caused by different strains. Predominance of one strain as described in the present study is seldom and strongly suggests that specific conditions resulted to exposure of teats to the same strain. From a practical point of view, this case confirms that reusable udder towels that are hand washed are a risk factor even if soaked in disinfectant solution between two milking times.

## Figures and Tables

**Figure 1 animals-11-00279-f001:**
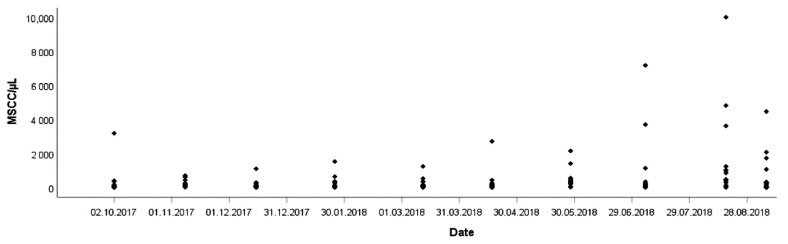
Increase of MSCC from July until September 2018 (data from individual monthly test day SCC).

**Figure 2 animals-11-00279-f002:**
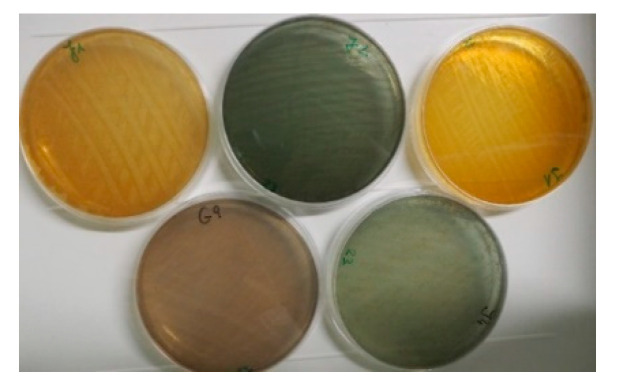
Colony morphology (pigmentation) of five *P. aeruginosa* strains belonging to the same MLVA type. All 54 isolates (32 from milk, 11 from the teat cleaning solution and eleven from the udder wipes) were identified as *P. aeruginosa* by MALDI-TOF MS. All score levels ranged between 2.11 and 2.45, indicating a reliable species identification. The species-specific PCR yielded an amplicon of the expected size (391 bp). With the various applied methods (ERIC-, REP-, BOX and RAPD-PCR, MLVA) all 54 isolates were indistinguishable.

**Table 1 animals-11-00279-t001:** Primer pairs used for MLVA.

Primer Pairs	Sequence
ms77L	5′-GCGTCATGGTCTGCATGTC-3′
ms77R	5′-TATACCCTCTTCGCCCAGTC-3′
ms127L	5′-CTCGGAGTCTCTGCCAACTC-3′
ms127R	5′-GGCAGGACAGGATCTCGAC-3′
ms142L	5′-AGCAGTGCCAGTTGATGTTG-3′
ms142R	5′-GTGGGGCGAAGGAGTGAG-3′
ms172L	5′-GGATTCTCTCGCACGAGGT-3′
ms172R	5′-TACGTGACCTGACGTTGGTG-3′
ms211L	5′-ACAAGCGCCAGCCGAACCTGT-3′
ms211R	5′-CTTCGAACAGGTGCTGACCGC-3′
ms212L	5′-TGCTGGTCGACTACTTCGGCAA-3′
ms212R	5′-ACTACGAGAACGACCCGGTGTT-3′
ms213L	5′-CTGGGCAAGTGTTGGTGGATC-3′
ms213R	5′-TGGCGTACTCCGAGCTGATG-3′
ms214L	5′-AAACGCTGTTCGCCAACCTCTA-3′
ms214R	5′-CCATCATCCTCCTACTGGGTT-3′
ms215L	5′-GACGAAACCCGTCGCGAACA-3′
ms215R	5′-CTGTACAACGCCGAGCCGTA-3′
ms216L	5′-ACTACTACGTCGAACACGCCA-3′
ms216R	5′-GATCGAAGACAAGAACCTCG-3′
ms217L	5′-TTCTGGCTGTCGCGACTGAT-3′
ms217R	5′-GAACAGCGTCTTTTCCTCGC-3′
ms222L	5′-AGAGGTGCTTAACGACGGAT-3′
ms222R	5′-TGCAGTTCTGCGAGGAAGGCG-3′
ms223L	5′-TTGGCAATATGCCGGTTCGC-3′
ms223R	5′-TGAGCTGATCGCCTACTGG-3′
ms207L	5′-ACGGCGAACAGCACCAGCA-3′
ms207R	5′-CTCTTGAGCCTCGGTCACT-3′
ms209L	5′-CAGCCAGGAACTGCGGAGT-3′
ms209R	5′-CTTCTCGCAACTGAGCTGGT-3′

**Table 2 animals-11-00279-t002:** Characteristics of the dairy farm.

Variable	Description
Stalls, Bedding	Free stall barn, cubicles with straw-manure mattresses, heifers on rubber mattresses without bedding
Cleaness Barn, Cows	Clean and dry cubicles, clean udders and cows
Udder Health Control	Monthly performed SCC measures, further preventive control measures like CMT or bacteriological examinations not established
Milking Technique	Tandem milking parlor, three milking units without automatic initial stimulation or automatic cluster take-off
Udder Preparation	Strip cup not used, teat cleaning with microfiber towels that were hand washed after milking and soaked into fresh water with added disinfectant (Dermisan Plus, Hypred, Bornheim, G) between the 12–12 h milking routine, teat skin clean but still wet before cluster attachment
Milking Procedure and Teat Disinfection	Short (<1 min) udder stimulation, overmilking of some cows observed, post milking teat disinfection with a barrier teat dip with 3.5% lactic acid (LactiFence, DeLaval, Tumba, S)
CMT	Not performed regularly
Dry Cow Management	Drying off 6–8 weeks prior calving, blanket dry cow therapy with cloxacillin (Orbenin, Pfizer, NY, USA), Dry cows not separated from the milking herd

**Table 3 animals-11-00279-t003:** Results of bacteriological examination (BE), MSCC measurement and interventions of all cows infected with *P. aeruginosa (Pseud.)*.

			First BE		Herd Survey (October 2018)			Treatment Control		
Cow	Quarter	CMT		Culture	SCC/mL	Culture	Intervention	SCC/mL	Culture	Intervention
Bon	FL	+++		++ Pseud.	876,000	+++ Pseud.		361,000	Negative	Culling
Ale	FL	++		++ Pseud.	523,000	++ Pseud.		243,000	++ Pseud.	Culling
Sil	FR	+++		+ Pseud.	21,000	Negative	Enrofloxacin	6000	Negative	Control
Sil	RL	+++		+ Pseud.	1,026,000	Negative		15,000	Negative	
Tam	FR	+++		++ Pseud.	n.d *	Negative	Enrofloxacin	1,523,000 *	Negative	Culling
Tam	RR			Negative	34,000	+ Pseud.		n.d.	Negative	
Tam	FL	+++		++ Pseud.	2,618,000 *	+ Pseud.		468,000	Negative	
Tam	RL	+++		+ Pseud.	10,000	Negative			Negative	
Sab	RR	+++		++ Pseud.	180,000	+ Pseud.		619,000	+ Pseud.	Culling
Sab	FL	++		++ Pseud.	437,000	+ Pseud.		148,000	+ Pseud.	
Sab	RL	+++		+ Pseud.	4,280,000	+ Pseud.		216,000	+ Pseud.	
Fil	RR	+++		++ Pseud.	663,000	Negative	Enrofloxacin	235,000	Negative	Culling
Fil	FL	+++		Negative	413,000	++ Pseud.		n.d.	+++ Pseud.	
Fil	RL	+++		Negative	1,962,000 *	Negative		2,703,000	+ Pseud.	
Flie	RL	+++		++ Pseud.	1,936,000 *	Negative	Enrofloxacin	1,888,000	+ Pseud.	Culling
Ali	FL	+++		+ Pseud.	1,089,000	++ Pseud.		4,322,000	++ Pseud.	Culling
*Bet*	*RL*			*n.d.*		*Negative*		*1,871,000*	*+++ Pseud.*	*Culling*

* Clinical mastitis CMT = California Mastitis Test (+ = weekly positive score, apparent thickening, ++ = clearly positive score gel formation, +++ strongly positive score, firm gel formation in the center of the cup), Culture: + = 1 to 5 CFU, ++ = 6 to 10 CFU, +++ = more than 10 CFU).

## Data Availability

All relevant data are presented in the manuscript.

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
