# Peer review of "Tracing Mastitis Pathogens—Epidemiological Investigations of a Pseudomonas aeruginosa Mastitis Outbreak in an Austrian Dairy Herd"

_animals, 2021, doi:10.3390/ani11020279_

Round 1

Reviewer 1 Report

Overall, I think the argument of this paper reflects what is requested by the field. This is really important to better understand the source of infections when outbreaks occur. I have only minor comments on what could be adjusted and improved

Line 26: move the comma after the word “where”.

The present study describes an outbreak of Pseudomonas (P.) aeruginosa mastitis in a 20 cows dairy herd where, through genotyping of isolates, reusable udder towels were identified as the source of infection.

Line 28: Is it necessary to use the acronymous (BE)?

The sentence sounds better like this: Sampling the cows during three herd surveys and bacteriological examination showed that P. aeruginosa was isolated from nine cows with a total of 13 infected quarters.

Line 29: Remove “in pure culture”

Line 29-31: Mastitis occurred as a mild clinical or subclinical infection. P. aeruginosa was additionally isolated from a teat disinfectant solution, containing N-(3-aminopropyl)-N-dodécylpropane-1,3-diamine 1 as the active component, and microfiber towels used for pre-milking teat preparation.

Line 31-33: Somewhere it needed to be mentioned that also Antimicrobial susceptibility testing was performed. What did you mean by “bacteriological examination”?

Line 34: What do you mean by “randomly selected”?

Line 35: Add acronyms “MALDI-TOF” at the end of the sentence.

Line 36-37: This sentence is not clear. In the abstract, It is not necessary to mention the primers that were used.

Genotyping of the isolates was performed by rep-PCR, Randomly Amplified Polymorphic DNA (RAPD), and Multiple Locus Variable-Number Tandem Repeat Analysis (MLVA).

Line 37-39: confirmed? The sentence for the conclusion is not clear.

Results obtained in this study suggested that one single strain was responsible for the whole outbreak and that the transmission occurred by a contaminated teat cleaning solution as a source of infection.

Line 90: remove the word “success”.

Line 97: remove the word “briefly”.

Line 108: remove the acronyms (BE).

Line 108-151: Underline or make clearer which are the titles of the sub-paragraphs. In general, it can be clearer if the topics and the description of the method are separated.

Line 108 -109: The somatic cell count should be not considered as a bacteriological examination. Move this sentence above, following Line 107.

Line 120: Change subtyping as “molecular strain typing”.

Line 129: Change “BE” as “Bacteriological Examination”

Line 132-133: Consider rephrasing this sentence. For example:

When P. aeruginosa growth was detected in the environmental samples, (how many?) colonies were randomly selected and stored at - 80 °C until further characterization and strain diversity assessment.

Line 138: P. aeruginosa needs to be in italic

Line 141: What Maldi score values did you use? Add details about score values and their acceptable range.

Line 142: P. aeruginosa needs to be in italic

Did you extract the DNA from the same colonies also tested by Maldi? This is fundamental.

Line 144: Add details about DNA extraction and then mention what kit has been used (Company name, town, and country). Also, did you quantify the DNA? For strain typing analysis is critical. If so, please describe the quantification method used.

Line 151: P. aeruginosa needs to be in italic

Line 159: Be consistent in how you write the number of animals if written with numbers or words. Examples: from 14 to five animals.

Line 168-169: sentence not clear, what does it mean “elevated in almost one quarter of nine cows”. Maybe “elevated in almost one-quarter for each of the nine cows”?

What does it mean “Milk secretions of three quarters were with the addition of flakes”?

Maybe “Milk secretions of three quarters showed presence of flakes”?

Line 170-174: All this sentence needs references, and it would be great to describe this a little bit more in detail instead of being so vague.

Line 193: this cfu/ml value to which sample is associated with? Be more specific.

Line 205: 54 is the total of isolates from milk and the environment. Below it is reported 32 as isolates from IMI, please also add the number for the environmental one.

Line 215: P. aeruginosa needs to be in italic

Line 218: Change “BE” as Bacteriological examination

Line 225: P. aeruginosa needs to be in italic

Line 237-243: This sentence is hard to follow and to locate it in the context of the other information reported. Are you talking about your study result? Or about what reported in the literature?

P. aeruginosa isolates belonging to the same MLVA type might show variations in their biochemical reactions. In particular .........

Line 247- 252: Are you talking in general about mastitis or specifically about Pseudomonas?

Line 251: “large number of bacteria”, what does it mean? Here, there are other bacteria than Pseudomonas? Does this mean that the clinical signs can be associated with some other bugs?

This sentence is not clear; please try to rephrase it.

Line 259: P. aeruginosa needs to be in italic

Line 265: P. aeruginosa needs to be in italic

Line 267: P. aeruginosa needs to be in italic

Line 273: P. aeruginosa needs to be in italic

Line 276-278: Sentence not clear. Rephrase it.

Line 277: P. aeruginosa needs to be in italic

Line 281: P. aeruginosa needs to be in italic

Line 285: P. aeruginosa needs to be in italic

Line 286: Serratia (S.) marcescens needs to be in italic

Line 289: S. marcescens needs to be in italic

Line 308: “point of view” repetition

Reviewer 2 Report

The main research findings of this paper will be important for preventing PA mastitis which causes great economic damage for dairy herds, and provides the good information to dairy farmers. I think however that there are a few improvements that should be made before publication.

General comment

Since there are many pathogens that cause unusual and progressive increase of milk SCC, the whole picture of bacterial examination in this study should be described. Or this point could be replaced by showing the results of monitoring milk SCC after intervention. 

Minor comment

Check the spelling "Oktober" in Table3.
